# Neuroendocrine–Immune Regulatory Network of *Eucommia ulmoides* Oliver

**DOI:** 10.3390/molecules27123697

**Published:** 2022-06-08

**Authors:** Yi Zhao, De-Chao Tan, Bo Peng, Lin Yang, Si-Yuan Zhang, Rui-Peng Shi, Cheong-Meng Chong, Zhang-Feng Zhong, Sheng-Peng Wang, Qiong-Lin Liang, Yi-Tao Wang

**Affiliations:** 1Macau Centre for Research and Development in Chinese Medicine, State Key Laboratory of Quality Research in Chinese Medicine, Institute of Chinese Medical Sciences, University of Macau, Macao 999078, China; mc05809@umac.mo (Y.Z.); yc07549@umac.mo (D.-C.T.); bopengtcm@gmail.com (B.P.); mb95803@um.edu.mo (L.Y.); yb97506@um.edu.mo (S.-Y.Z.); mc05808@umac.mo (R.-P.S.); cmchong@um.edu.mo (C.-M.C.); zhangfengzhong@um.edu.mo (Z.-F.Z.); swang@um.edu.mo (S.-P.W.); 2MOE Key Laboratory of Bioorganic Phosphorus Chemistry & Chemical Biology, Beijing Key Lab of Microanalytical Methods & Instrumentation, Department of Chemistry, Center for Synthetic and Systems Biology, Tsinghua University, Beijing 100084, China

**Keywords:** *Eucommia ulmoides* Oliver (*E. ulmoides*), neuroendocrine–immune, cancer, network pharmacology, natural products

## Abstract

*Eucommia ulmoides* Oliver (*E. ulmoides*) is a popular medicinal herb and health supplement in China, Japan, and Korea, and has a variety of pharmaceutical properties. The neuroendocrine–immune (NEI) network is crucial in maintaining homeostasis and physical or psychological functions at a holistic level, consistent with the regulatory theory of natural medicine. This review aims to systematically summarize the chemical compositions, biological roles, and pharmacological properties of *E. ulmoides* to build a bridge between it and NEI-associated diseases and to provide a perspective for the development of its new clinical applications. After a review of the literature, we found that *E. ulmoides* has effects on NEI-related diseases including cancer, neurodegenerative disease, hyperlipidemia, osteoporosis, insomnia, hypertension, diabetes mellitus, and obesity. However, clinical studies on *E. ulmoides* were scarce. In addition, *E. ulmoides* derivatives are diverse in China, and they are mainly used to enhance immunity, improve hepatic damage, strengthen bones, and lower blood pressure. Through network pharmacological analysis, we uncovered the possibility that *E. ulmoides* is involved in functional interactions with cancer development, insulin resistance, NAFLD, and various inflammatory pathways associated with NEI diseases. Overall, this review suggests that *E. ulmoides* has a wide range of applications for NEI-related diseases and provides a direction for its future research and development.

## 1. Introduction

*Eucommia ulmoides* Oliver (*E. ulmoides*) is a monotypic genus *Eucommia*, also known as tuchong in Japanese and tu-chung in Korean, and was recorded in *Shen Nong Ben Cao*, a classic Chinese medical book [1]. It is characterized by being resistant to cold (−40 °C) and hot (44 °C) conditions [2]. Generally, *E. ulmoides* is cultivated in the southern area of Qingling in China, including the Guizhou, Sichuan, Hubei, Shaanxi, Hunan, Gansu, Yunnan, Anhui, Guangxi, Henan, Zhejiang, and Jiangxi provinces (Figure 1A) [3,4,5]. *E. ulmoides* has been utilized for at least 2000 years according to ancient Chinese medical records [6]. In 1955, the first global conference on the pharmacological effects of *E. ulmoides* was organized in Leningrad, and the scientists there suggested that *E. ulmoides* was effective in decreasing blood pressure [7]. From then on, *E. ulmoides* has attracted intensive attention worldwide and a great deal of scientific research and clinical trials have been undertaken to study its biological functions and pharmacological effects (Figure 1B).

Recently, the barks and leaves of *E. ulmoides* have been listed separately in the *Chinese Pharmacopoeia* (2020 version), with different quality control criteria, but identical functions, including nourishing the liver and kidney, and strengthening bones and muscles [8]. Additionally, *Eucommia ulmoides* Oliver barks (EUE) were recorded in the *European Pharmacopoeia* (9th Edition), *Japanese Pharmacopoeia* (17th Edition, English Version), *Hong Kong Chinese Materia Medica Standards* (Volume 3), and *Taiwan Herbal Pharmacopeia* (3rd Edition). In particular, among the 200 standard formulations in the *Taiwan Herbal Pharmacopeia*, 5 prescriptions involve EUE [9]. Traditionally, *E. ulmoides* has been considered to have the properties of tonifying the kidney and liver, strengthening bones and muscles, and fixing meridians from ancient records [6]. While in modern pharmacology, 204 chemical constituents have been identified from the leaves, barks, seeds, and flowers of *E. ulmoides*, and these compounds are divided into phenols, iridoids, lignans, flavonoids, terpenoids, sterols, gutta-percha, polysaccharides, unsaturated fatty acids, amino acids, and mineral elements [10]. Several components display vital biological functions both in vivo and in vitro, such as hypolipidemic, antihypertensive, antidiabetic, anti-inflammatory, antioxidative, neuroprotective, hepatoprotective, bone-metabolic, renoprotective, anti-aging, anti-fatigue, antidepressant, hypnotic-sedative, immune regulation, cognitive improvement, uterine smooth relaxation muscles, and erectile function enhancement [11,12,13,14,15]. Apart from being a medicine, *E. ulmoides* has also been employed as a health supplement popular in China, Japan, and Korea [16]. In short, *E. ulmoides* has great economic value and the potential for being used in novel drugs.

The neuroendocrine–immune (NEI) regulatory network (Figure 2) is a complex system, incorporating the nervous system, endocrine system, and immune system, to maintain homeostasis and plays a pivotal role in the treatment of complicated diseases, with the involvement of bioregulatory signals such as neurotransmitters, hormones, and cytokines/chemokines [17,18,19]. This review aims to systematically summarize the chemical components, biological activities, and pharmacological effects of *E. ulmoides* on NEI diseases, which will provide a reference for research, development, and application of the active components of *E. ulmoides*.

## 2. Neuroendocrine–Immune Regulatory Network

Besedobsky first proposed the NEI network in 1997. The interactions among the NEI system represent a complete communication circuit by sharing common signaling ligands and their receptors. In general, the nervous system regulates the immune system in two ways. One is through the release of neurotransmitters or neuropeptides such as acetylcholine, 5-HT, and opioid peptides from the endings of autonomic nerves, which act on immune cells and organs (bone marrow, thymus, lymph nodes, and gut) [20]. The second regulatory mechanism is through the hypothalamic–pituitary–adrenocortical (HPA), hypothalamic–pituitary–thyroid (HPT), hypothalamic–pituitary–gonadal (HPG), and hypothalamic–pituitary–somatotropic (HPS) axes to regulate the immune system. The release of neurohormones such as corticotropin-releasing hormone (CRH) from the paraventricular nucleus of the hypothalamus can stimulate the anterior pituitary gland to secrete adrenocorticotropic hormone (ACTH). ACTH acts on the adrenal cortex and promotes the secretion of glandular hormones (e.g., glucocorticoids) to mediate the immune response [21]. On the other hand, immune cells can generate various immune mediators to trigger the activation of the nervous system in response to inflammatory and invasive stimuli. For example, IL-1 can upregulate the secretion of CRH [22], IL-1β, and TNF-α, which are regarded as potential neurotoxic substances [23], while the proinflammatory cytokine induction of interferon-γ (IFN-γ) provides neuroprotection during acute neuroinflammation by inducing the secretion of IL-6 [24].

The secretion of hormones also has a role in the nervous system. For example, thyroxine is important for the development of the brain [25], oxytocin can improve the learning and memory of mice by regulating the hippocampus [26], and vasopressin can also enhance memory [27]. As for the immune system, hormones such as glucocorticoids secreted by the adrenal cortex have both anti-inflammatory and proinflammatory effects [28]. As gonadal hormones, both androgens and estrogens can improve immunity [29,30]. In particular, estrogen can alter the immune response by binding to specific receptors on immune cells, resulting in their proliferation, and they play a feedback regulatory role on the hypothalamic–pituitary axis [29,31].

The NEI network is involved in multidirectional functions and multiple systems. Once the imbalance or alteration of the NEI network occurs, various diseases may follow, such as multiple sclerosis, fatigue, inflammation, psoriatic arthritis, systemic lupus erythematosus, depression, anxiety, cancer, and obesity [32,33,34,35,36,37,38,39,40,41]. From a holistic perspective, *E. ulmoides* has a “multi-components, multi-targets” profile for the treatment of various diseases. Hence, it is valuable to understand the pharmacological effects of *E. ulmoides* in the NEI network and fill this research gap.

## 3. Chemical Compounds of *E. ulmoides*

The chemical profile of *E. ulmoides* is described as a mixture of iridoids, lignans, flavonoids, phenolics, steroids, terpenoids, polysaccharides and other compounds (Figure 3). It is accepted that all organic tissues of *E. ulmoides* contain similar kinds of chemical substances, however, various components such as phenolics are found predominantly in leaves and iridoids in barks [6]. Lignans in *E. ulmoides* are accumulated in barks with rich glycosides [10]. As a notable component of lignans, (+)-pinoresinol di-*O*-β-D-glucopyranoside is acknowledged as one of the quality control factors of EUE in the *Chinese Pharmacopeia*. The content of (+)-pinoresinol di-*O*-β-D-glucopyranoside should be higher than 0.1% in EUE, while chlorogenic acid should be higher than 0.08% for *Eucommia ulmoides* Oliver leaf (EUL) according to the *Chinese Pharmacopeia* (2020 vision) [8]. Chlorogenic acid possesses good medical translational potentiality with anti-obesity, anti-diabetes, anti-hypertension, anti-metabolic syndrome, and anti-microbiota ability [42]. The phenolic components of *E. ulmoides* play an important role, accounting for the anti-oxidant capacity of *E. ulmoides* and attributed to their hydroxyl groups. Another leaf-particular compound is flavonoid, its content is a key criterion for determining the quality of raw herbs and products of *E. ulmoides* [7]. Iridoids are abundant in the barks, leaves, flowers, and seeds of *E. ulmoides* an include GPA, geniposide, and aucubin [43]. Several factors, such as the location of growth, climate, harvest method, and material processing procedures, affect the accumulative content of iridoids [6,44].

Polysaccharides are the most abundant components of *E. ulmoides*, with immune-related effects. It has been shown that *E. ulmoides* polysaccharides effectively reduce the release of inflammatory factors to achieve anti-inflammatory effects in both in vitro and in vivo models [13]. Additionally, *E. ulmoides* polysaccharides containing galacturonic acid serve as novel biomaterials, with a binding affinity for growth factors and platelet-derived growth factor-BB (PDGF-BB), promoting tissue regeneration [45]. In addition to PDGF-BB, *E. ulmoides* polysaccharides also bind to fibroblast growth factor-2, which is a pro-angiogenic growth factor. In general, *E. ulmoides* polysaccharides could potentially be used as a new approach to promote angiogenesis [46]. Meanwhile, the double bond and hydroxyfuran ring structure is easily altered, which is responsible for the blackening of *E. ulmoides* during herbal material processing. Other compounds, including sterols, triterpenes, carbohydrates, antifungal proteins, fatty acids, vitamins, and amino acids, are also compounds of *E. ulmoides* [47]. Among these, gutta-percha has filamentous flexibility and wide applications both in the material industry and medical field. Gutta-percha consists of trans-1,4-polyisoprene, which has unique features such as rubber–plastic duality, abrasion resistance, and slip resistance. Therefore, it has high economic value and translational potential [10].

## 4. Pharmacological Effects of *E. ulmoides* on NEI Network-Associated Diseases

The NEI network mediators and their end products have widespread effects at the systemic and cellular levels. They are responsible for disease behavior, such as cancer, neurodegenerative diseases (Alzheimer’s disease (AD) and Parkinson’s disease (PD) which are explained in Figure 4A), metabolic disorders including obesity, insulin resistance, diabetes, cardiovascular disease, and dyslipidemia, as well as osteoporosis, fatigue, depression, and anxiety [32,33,34,35,36,37,38,39,40,41,48,49,50] (Figure 4B). Thus, maintaining the balance of the NEI network may bring benefits for the treatment of these diseases. Through reviewing the literature on *E. ulmoides*, we found that the herb does affect these diseases, as discussed in the following sections and summarized in Table 1.

### 4.1. Cancer

Cancer is a complex disease that presents as a disorder of the NEI system and is regulated at systemic, organ, and microenvironmental levels. At each level, corresponding signaling factors such as neurotransmitters (epinephrine, norepinephrine, Ach, and 5-HT), hormones (estrogens, androgens, prolactin, insulin, glucocorticoids, and prostaglandins), and cytokines (IL-6, TNF-α, and BDNF) are transmitted along the NEI axis, thereby inducing the proliferation and invasion of carcinoma cells, and ultimately contributing to cancer occurrence or progression [32,94,95,96,97,98]. Several studies have revealed the extensive anti-tumor effects of *E. ulmoides* on a variety of tumor cells, such as gastric cancer (AGS) [51], colon cancer (HCT116, LoVo, SNU-C4) [56,57,59,60], cervical cancer (Hela) [55,57], breast cancer (MDA-MB-231, T47D) [55], lung cancer (LLC, A549) [57,58], liver cancer (H_22_) [53], and glioblastoma (U251, U87, HS683, A172) [52].

*E. ulmoides* extract, in the presence of a nano bamboo charcoal drug delivery system, which promptes the adsorption and release capacity of *E. ulmoides* extract, has shown inhibitory effects on the colon cancer cell line HCT116 [59]. Acetone extract from EUL is cytotoxic towards lung cancer cells (A549), colon cancer cells (SNU-C4), and cervical cancer cells (Hela), with IC_50_ values of 53.4, 53.8, and 88.3 μg/mL, respectively [57]. Aside from extracts, several specific compounds of *E. ulmoides* have also demonstrated promising anti-cancer effects. For instance, chlorogenic acid effectively inhibited the proliferation of gastric cancer cell AGS in vitro, with IC_50_ values of 0.73 and 0.31 mg/mL for 24 and 48 h, respectively [51]. Chlorogenic acid also inhibited the proliferation of and induced apoptosis in the colon cancer cell lines HCT116 and LoVo [56]. Meanwhile, EUL extract and chlorogenic acid suppressed the invasion and migration of HCT116 and LoVo cell lines [60]. Eucommicin A, a β-truxinate-type dimer of chlorogenic acid identified from EUL, has been shown to display a selective inhibitory effect on cancer stem cells and tumor sphere formation [54]. Another three pentacyclic triterpenoids isolated from *E. ulmoides*, namely betulinic acid, lupeol, and 3-O-laurylbetulinic acid, inhibited the growth of Hela, MDA-MB-231, and T47D cells. Moreover, these three compounds also effectively induced apoptosis and mitochondrial fragmentation, and suppressed the lysosome production in Hela cells [55]. In addition, the total polysaccharides of *E. ulmoides* may inhibit the proliferation of cancer cells via upregulation of caspase-3 expression in the lung cancer cell line LLC [58]. The total flavonoids from *E. ulmoides* suppressed the proliferation, migration, and invasion of glioblastoma cells and sensitized glioblastoma to radiotherapy, causing a notable reduction in the number of normalized colonies [52]. The potential mechanism of radiosensitization was achieved by decreasing the ratio of Bcl-2/Bax, inducing apoptosis in glioblastoma cells, and downregulating the expression of HIF-1α, MMP-2, and Wee1 via the HIF-1α/MMP-2 signaling pathway [52]. Similarly, the total flavonoids of *E. ulmoides* inhibited tumor growth in H_22_ tumor-bearing mice through an increase in Bax expression and a decrease in Bcl-2 expression [53]. Studies have shown that *E. ulmoides* has cytotoxic and anti-tumor activities, however, there is limited systemic research on its anticancer effects and mechanisms in vitro and in vivo. Therefore, further investigation into these factors is required.

### 4.2. Alzheimer’s Disease (AD)

AD is the most common neurodegenerative disease, it is characterized by a progressive decline in memory and cognitive impairment [99]. Neuroinflammation is a common feature, involving neuronal loss in the AD brain. The disturbance of the neuroendocrine system has been associated with cognitive decline in AD patients. Thus, the dysregulation of homeostasis of the NEI system is implicated in the development of AD.

Active microglia are involved in neuroinflammation in AD. In lipopolysaccharide (LPS)-stimulated BV-2 microglia (mouse, C57BL/6, brain, microglial cells), ethyl acetate extract of EUE could reduce two inflammatory mediators, nitric oxide (NO) and prostaglandin E_2_ (PGE_2_) by inhibiting the expression of inducible NO synthase (iNOS) and COX-2. The activities were found to suppress the release of inflammatory molecules, including TNF-α and IL-1β. In addition, these effects were associated with suppressing the phosphorylation of MAPKs and PI3K/Akt as well as glycogen synthase kinase-3β (GSK-3β), thereby inhibiting NF-κB activation and inducing Nrf2-dependent heme oxygenase (HO)-1 activation [61]. This study reveals that *E. ulmoides* has components with anti-neuroinflammatory activity. The potent anti-AD capacity of EUE water extracts in hydrogen peroxide (H_2_O_2_)-induced SH-SY5Y cells (a thrice-subcloned cell line derived from the SK-N-SH neuroblastoma cell line) is attributed to its ability to enhance cell viability, inhibit cytotoxicity and DNA condensation, ameliorate reactive oxygen species (ROS) production, reduce mitochondria membrane potential (MMP), and regulate the expression of Bcl-2 and Bcl-xL. Additionally, it inhibits the release of cytochrome c from mitochondria to the cytosol and attenuates JNK, p38 MAPK, ERK 1/2, and PI3K/Akt phosphorylation [63]. In AD mice with scopolamine-induced learning and memory impairments, EUE improved memory by enhancing or protecting cholinergic signaling. EUE markedly suppresses the activities of acetylcholinesterase (AChE) and thiobarbituric acid reactive substances (TBARS) in the hippocampus and frontal cortex, and elevates BDNF and scopolamine-induced phosphorylation of cAMP element-binding protein (CREB) in the hippocampus of mice [64].

Geniposidic acid (GPA) is a compound found in EUE. It has been reported that GPA-administrated APP/PS1 mice had improved spatial learning and memory functions, and attenuated amyloid-β (Aβ) plaques. This performance was primarily due to the suppressing activation of astrocytes and microglia, downregulation of pro-inflammatory cytokines and iNOS, and upregulation of anti-inflammatory cytokines and Arg-1. Moreover, GPA reduces the expression of HMGB-1 receptors (TLR2, TLR4 and RAGE) and mediates MyD88 and the expression of AP-1 and NF-κB family members (c-Fos, c-Jun and p65), serving as a potential drug candidate through reducing Aβ deposition and neuroinflammation to reverse pro-inflammatory states. Thus, it can be used as an auxiliary therapy of AD [67]. Macranthoin G (MCG), a derivative of methyl chlorogenic acid, protects against H_2_O_2_-induced cytotoxicity in rat PC12 cells [68]. Here, the potential mechanism is that MCG effectively stabilizes MMP and enhances the antioxidant enzyme activities of superoxide dismutase (SOD), catalase, glutathione peroxidase, and intracellular glutathione (GSH). Furthermore, it reduces malondialdehyde (MDA) levels, intracellular ROS, caspase-3 activation and apoptosis [68]. In addition, MCG treatment mimics the cellular damage from H_2_O_2_ by downregulating the NF-κB signaling pathway and activating phosphorylation of IκBa, p38, and ERK [68].

### 4.3. Parkinson’s Disease (PD)

PD ranks as the second most common neurodegenerative disease and is caused by the progressive degeneration of dopaminergic neurons in the substantia nigra pars compacta of the midbrain [100]. Neuroendocrine abnormalities, such as decreased dopamine levels and the disruption of melatonin secretion, are common pathogenic factors of PD [100]. In addition to this, an imbalance in the HPA axis and increased cortisol and serum levels of TNF-α present as clinical features in PD patients, suggesting that the NEI system plays a key role in PD progression.

Recently, it was reported that EUE reduces a series of PD-related events such as mitochondrial dysfunction, ROS increase, dopamine decrease, neuronal death, and motor behavior defects. For example, pre-treatment with EUE ethyl acetate extracts on SH-SY5Y cells significantly reduced PD-related neurotoxin 6-hydroxydopamine (6-OHDA)-induced cell death and cytotoxicity. The underlying mechanism may involve inhibition of ROS generation and activation of JNK, Akt, and GSK-3β, thereby leading to the suppression of apoptotic cascade and NF-κB translocation [62]. An in vivo study on male C57BL/6J mice showed that EUE extracts mediated anti-inflammatory and anti-PD effects through downregulating the expression of p38 MAPK, JNK, and FOS-like antigen-2 (Fosl2) [65]. Also, three-day pretreatment with EUE reduced striatal neurotransmitter loss and alleviated motor abnormalities in 1-methyl-4-phenyl-1,2,3,6-tetrahydropyridine (MPTP)-treated PD mice. The five compounds identified from EUE extracts, including betulin, wogonin, oroxylin A, GPA, aucubin, and lignans, can rescue the 1-methyl-4-phenylpyridinium (MPP^+^)-induced disorder of protease activity and MG132-induced cytotoxicity in SH-SY5Y cells [66]. Another study using C57BL/6J mice as a model, also induced with MPTP, found that the EUE water extracts possessed anti-neuroinflammatory effects also through downregulating the expression of p38 MAPK, JNK, and Fosl2. Apart from EUE extracts, EUL extracts have similar therapeutic effects in PD models. It was found that EUL reversed the loss of dopaminergic neurons and relieved motility disorders in MPTP-induced PD zebrafish via activating autophagy to facilitate α-synuclein degradation. Furthermore, this finding was confirmed by molecular docking, which revealed the association between autophagic factors (Pink1, Beclin1, Ulk2, and Atg5) and phenolic acid [101].

Taken together, the effects of *E. ulmoides* on neuroprotection are related to the following mechanisms: the suppression of AChE activity and production of inflammatory mediators such as NO and PGE_2_; the attenuation of intracellular ROS accumulation; the downregulation of the expression of p38, JNK, and Fosl2; the enhancement of cholinergic signaling; the promotion of 5-HT release through enhancing synapsin I expression; the attenuation of gliosis and regulation of neurotransmission; the induction of autophagy and inhibition of necroptosis; and the alleviation of Aβ deposition and neuroinflammation.

### 4.4. Hyperlipidemia

Hyperlipidemia is the status of elevated lipid levels in the blood, which includes high levels of low-density lipoproteins (LDL), total cholesterol (TC), triglyceride (TG), or lipoproteins and a relatively low level of high-density lipoprotein (HDL) [102,103]. Vascular endothelial cells are essential in maintaining vascular homeostasis and play a key role in the NEI system [104,105,106,107,108,109]. Endoplasmic reticulum (ER) stress in endothelial cells is one of the causative factors of hyperlipidemia because it leads to increased oxidative stress or inflammation, thereby resulting in cell death [110]. Hence, ER stress has a tight connection with the NEI system.

It was found that a number of active compounds abundant in *E. ulmoides*, including asperuloside, GPA, quercetin, chlorogenic acid, and aucubin, possess strong lipid lowering effects [70,111]. In addition, EUL has the potential to control dyslipidemia in non-alcoholic fatty liver disease. EUL supplementation (200 mg/kg) promotes recovery from high-fat diet (HFD)-induced lipid dysmetabolism accompanied by inhibiting ER stress and enhancing lysosomal functions, thereby increasing autophagic flux by suppressing the mTOR-ER stress signaling pathway [69]. Moreover, EUL and its two active constituents, aucubin and geniposide, inhibit palmitate-induced ER stress by increasing lysosomal enzyme activity, and reduce liver lipid accumulation by secreting apolipoprotein B, associated triglycerides, and cholesterol in human HepG2 cells [70]. Another explanation is that the enhancement of lysosomal activity is achieved by regulating the translocation of lysosomal Bax [72]. Carbon tetrachloride (CCl_4_) decreases GSH and increases MDA accompanied by activated P450 2E1 in rat liver, whereas pretreatment with EUE extracts could significantly reduce the deleterious effects of CCl_4_ such as the ER stress response and ROS increase, which are regarded as a possible mechanism in the anti-dyslipidemic effect of EUE extracts [71]. Hao et al. found that the chlorogenic acid in EUL exhibited anti-dyslipidemia and anti-obesity efficacy via regulation of multiple molecules. Treatment with CAEF at a concentration of 25 mg/L for 48 h promoted the lipid droplets excreted from HepG2 cells and elevated the TG and TC efflux by upregulating ABCA1 and CYP7A1. Moreover, CAEF inhibited the level of endogenous HMG-CoA reductase and SREBP2 [73]. Previous studies have revealed that EUL has beneficial effects against the progression of metabolic syndrome and can promote fatty acid oxidation. More specifically, quercetin, a major flavonoid component of EUL, activates the PPAR signaling pathway leading to a hypolipidemic action, indicating that EUL can be used safely as a functional food for the prevention of metabolic syndrome [112]. The total flavonoids in EUL can dramatically lower the serum levels of cholesterol, triglyceride, lipoprotein, apolipoprotein, and density lipoprotein cholesterol, and, remarkably, can increase the levels of HDL, cholesterol, and apolipoprotein A [74].

The impact of *E. ulmoides* on lipid metabolism is associated with the following mechanisms: reinforcement of lysosomal activity leading to decreased ER stress; enhancement of lipid metabolism through activating adenylate-activated AMPK; inhibition of HMG-CoA reductase; activation of the PPAR signaling pathway; alleviation of ROS regulation; and increase of hyperdensity lipoprotein cholesterol and apolipoprotein A.

### 4.5. Hypertension

Hypertension is the status of sustained higher than normal blood pressure [113]. As an important system in the regulation of blood pressure, the renin–angiotensin–aldosterone system (RAAS) not only regulates blood pressure and fluid homeostasis [114], but also plays an important modulating role in immune function [115]. In RAAS, renin can convert angiotensin (Ang) to Ang I, and Ang convertase (ACE) catalyzes the conversion of Ang I to Ang II. Although Ang II has two receptor subclasses, the Ang II type 1 receptor (AT1R) and the Ang II type 2 receptor, AT1R mediates the primary action of Ang II [116,117]. Ang II activation enhances the generation of inflammatory cytokines through the NF-κB signaling pathway [115], whereas inhibition of Ang II reduces the production of TNF-a, IL-1, IL-6, and IL-18 in arthritis models [118,119,120].

The extract of the male flower of *E. ulmoides* was reported to reduce blood pressure by regulating ACE2–Ang-(1-7)–Mas signaling pathway in spontaneously hypertensive rats [82], and EUL extract prevented hypertension in addition to aortic media hypertrophy [11]. Megastigmane glycosides are isolated from *E. ulmoides* and exhibit moderate inhibitory effects on ACE and the modulation arterial blood pressure [121]. In addition, the combination of *E. ulmoides* and *Tribulus terrestris* showed anti-hypertensive effects in spontaneously hypertensive rats, possibly by regulating intestinal microbiota and beneficial metabolites [122]. Some marketed formulas containing *E. ulmoides*, such as Duzhong Jiangya tablets and Quan-*E. ulmoides* capsules, have been used clinically for the treatment of hypertension [123,124].

### 4.6. Diabetes Mellitus (DM)

Diabetes mellitus (DM) is a chronic disorder related to glucose metabolism and the secretion of insulin, with serious clinical complications resulting from the damage of macrovascular and microvascular blood vessels [125]. Insulin production is closely related to the HPA axis, a vital feedback regulatory system in the NEI network [126]. In the pituitary gland and plasma of diabetic rats, the ACTH secreted by the pituitary gland was higher than that of the normal group [127]. Elevated ACTH promoted the secretion of glucocorticoids by the adrenal cortex, which acted as antagonists of insulin [128]. This led to a decrease in insulin content, triggering glucolipid disorders and promoting the development and progression of diabetes [129].

Several studies have confirmed that *E. ulmoides* can exert anti-diabetic effects by decreasing the level of plasma glucose and regulating the activity of SOD and MDA [83]. In addition to the EUE, EUL has become a promising agent in the treatment of DM [84]. In rats fed with an HFD, both the extract and powder of EUL improved insulin resistance in a dose-dependent manner and decreased plasma glucose levels [84]. In addition, asperuloside, a compound of EUL, can reduce plasma glucose in rats fed an HFD by promoting insulin sensitivity, implying that *E. ulmoides* is involved in the regulation of glucose and insulin [85].

The secondary complications of DM are more severe, leading to adverse outcomes [130]. Combined treatment with quercetin and crocin isolated from *E. ulmoides* can decrease the level of fasting blood glucose and accumulation of lipids, and improve renal fibrosis in vivo [92]. It was reported that effervescent granules with 5% chlorogenic acids and 5% polysaccharides isolated from EUL and moso bamboo leaves boosted glucose uptake to 156.35% in high-glucose cultured HepG2 cells [86]. This activity likely depends on the inhibitory effect on glucose-6-phosphate displacement enzyme and α-glucosidase [86]. Moreover, the glucose uptake induced by effervescent granules was similar to those of metformin and insulin under this condition [86]. *E. ulmoides* also prevented the progression of diabetic nephropathy. Notably, in another study with streptozotocin-induced type 1-like DM rats, *E. ulmoides* reduced urea nitrogen and creatinine without altering the level of blood glucose [87]. In the model of streptozotocin-induced DM mice, oral administration of *E. ulmoides* extract at a concentration of 200 mg/kg for 6 weeks suppressed the formation of advanced glycation end products (AGEs) and receptors for AGEs that were related to the renal damage [88]. Compared with the bark and root of *E. ulmoides*, EUL containing isoquercetin, 6″-O-acetyl-astragalin, rutin, and astragalin have potent inhibitory effects on the formation of N(ε)-(carboxymethyl)lysine and N(ω)-(carboxymethyl)arginine, which accumulate in lens crystallins [89]. The lignans existing in *E. ulmoides* inhibited the damage of retinal endothelial cells caused by AGEs in vitro [16]. These components also targeted Ang II, which elevated the proliferation and protein levels of Col I, Col III, and Col IV [90].

### 4.7. Obesity

According to the guidelines of the World Health Organization (WHO), obesity is defined as abnormal or excessive fat accumulation and poses a great threat to public health [131]. Obesity is associated with dysfunction of the HPA axis. There is a difference in the secretion and activity of cortisol in obese and normal weight individuals, suggesting that dysregulated HPA axis activity is present in obese patients [132,133]. High levels of cortisol causes abnormal adipose tissue metabolism, which leads to obesity [134]. Disruption of the neuroendocrine mechanisms that regulate CRH and ACTH release have been repeatedly shown to lead to abdominal obesity in both men and women [135]. There is also growing evidence that in humans, the phenotype of abdominal obesity may be characterized by an overactivation or overreaction of the HPA axis [136].

Asperuloside, an iridoid glycoside isolated from *E. ulmoides*, decreased body weight and improved insulin resistance in HFD-fed mice via improvement of gut microbiota [91]. In comparison with the control group, HFD-fed mice treated with asperuloside showed no difference in body weight, while increased body weight was observed in the mice treated with an HFD only, indicating that asperuloside can prevent obesity without altering daily intake of food [91]. Interestingly, both EUL aroma and EUL extracts exhibited anti-obesity activity in rats by elevating the mRNA levels of Cpt2, Acad, complex II, and complex V in the liver, and increasing the expression of BDNF, Akt, and phospholipase Cγ in the hypothalamus [93]. Hosoo et al. confirmed the anti-obesity and anti-hypertensive effects of EUL extract, and suggested that the active ingredients were GPA and asperuloside [11]. Compared with the HFD-induced model mice, EUL-treated mice had lower body weight and blood pressure and improved plasma adiponectin/leptin ratio [11]. In addition, the chlorogenic-acid-enriched EUL extract also prevented obesity and dyslipidemia, which might be involved in activating AMP-activated protein kinase (AMPK) and suppressing SREBP2 and HMGCR signaling pathways [73]. The chlorogenic-acid-enriched EUL extract also inhibited TC and TG production in the liver by regulating cholesterol synthesis genes. This extract can increase the expression of both ABCA1, which is responsible for cholesterol efflux from cells and absorption from the intestinal tract, and CYP7A1 to enhance bile acid and TC secretion and improve lipid metabolism in HepG2 cells [73].

### 4.8. Osteoporosis

Osteoporosis is a systemic skeletal disease characterized by reduced bone mass and microarchitectural degeneration that may increase bone fragility and risk of fracture, leading to severe complications [137]. The etiology of osteoporosis is related to endocrine and metabolic disorders, as the microenvironment produced by osteoblasts can condition the immune system to regulate the development of bone metabolism through B cells, T cells, dendritic cells, and multiple cytokines [138,139,140,141]. A break in the balance between the abnormally activated osteoblasts and osteoclasts of the immune system can lead to osteoporosis [142,143,144]. Furthermore, endogenous glucocorticoids play an essential role in bone homeostasis [145], as glucocorticoids can inhibit bone formation by prolonging the lifespan of osteoclasts and promoting osteoblast apoptosis [146,147,148]. In addition, insulin-like growth factor 1, which promotes bone formation by stimulating type I collagen synthesis, also inhibits collagen degradation and osteoblast apoptosis [149]; however, its gene transcription is suppressed by glucocorticoids [150,151]. Recent studies have demonstrated that the prevention and treatment of osteoporosis includes three aspects: increasing estrogen levels, regulating cytokines associated with bone metabolism, and promoting the production of osteoprotective hormones [152].

*E. ulmoides* has been utilized to treat fractures and bone disorders for a long time [10,153]. It has been proved that the extract from *E. ulmoides* seed, with cyclic ether terpenes as the main component, significantly increases bone density of femoral trabeculae, as well as the content of calcium, phosphorus, strontium, zinc, magnesium, and iron in the femur, and effectively prevents osteoporosis caused by testicular resection [154,155]. In addition, the total glycosides of *E. ulmoides* seed can apparently boost serum alkaline phosphatase (ALP) levels and can strengthen bone formation to treat osteoporosis in men without affecting serum androgen levels [156]. In addition, EUE extract prevented estrogen-deficiency-induced bone loss and deterioration of the trabecular bone structure, thereby maintaining bone biomechanical capacity, and preventing postmenopausal osteoporosis and obesity [6]. EUE extract also dose-dependently inhibited the ovariectomy-induced decrease in total bone mineral density of the femur and decreased levels of the bone conversion index osteocalcin, alkaline phosphatase, and deoxypyridoxine, as well as urinary calcium and phosphorus excretion levels [157,158]. Both in vivo and in vitro studies have shown that total lignans extracted from EUE effectively suppressed the loss of bone mass induced by ovariectomy and inhibited osteoclastogenesis by increasing osteoprotegerin and reducing the expression of NF-κB ligands [75]. In addition, EUE extracts induced growth hormone release and regulated bone maturation and remodeling [10]. Saline-formulated EUE was able to increase the activity of serum ALP and IGF-1 to treat senile osteoporosis through reducing calcium content in the blood and simultaneously converting calcium ions into precipitates [159]. It was reported that the signaling pathways related to the differentiation of bone marrow mesenchymal stem cells into osteoblasts could be induced by *E. ulmoides* [153] by activating the Wnt/β-catenin [160], MAPK/ERK [161,162], and RhoA/ROCK [163] signaling pathways. Therefore, it is believed that EUE extracts can be used to prevent and treat osteoporosis. Aside from its extracts, several active components of *E. ulmoides* such as lignans, cyclic enol ether terpenes, flavonoids, and phenylpropanoids, are involved in the proliferation and differentiation of osteoblasts, osteoclasts, and medullary mesenchymal stem cells by regulating estrogen levels and the expression of bone-metabolism-related cytokines and osteopontin, thus effectively preventing osteoporosis [156].

### 4.9. Insomnia

Insomnia is a disorder characterized by dissatisfying sleep quantity and quality, and is closely related to hyperfunction of HPA and increasing activity of CRF [164]. Intravenous CRF may result in diminished growth hormone secretion and reduced slow-wave sleep in normal men [165]. Findings in healthy subjects suggested that experimentally induced elevations in ACTH and CRF disrupted sleep [166]. The water-soluble alkaloids in male *E. ulmoides* flowers have a great sedative–hypnotic effect, and they effectively reduced spontaneous activity, prolonged sleep time, and shortened sleep latency. Additionally, the eclamptic ratio caused by nikethamide was reduced and the latent period of convulsions was prolonged in mice [167]. Eucommiol, a monomeric compound extracted from male *E. ulmoides* flowers, exhibited several effects on mice, including a decrease in spontaneous activity and prolongation of sleep duration [80]. In addition, studies revealed that astralagalin, another ingredient isolated from *E. ulmoides*, exhibited significant sedative–hypnotic activity in mice, including decreasing spontaneous activity, attenuating sleep latency, and lengthening sleep duration [81]. In summary, eucommiol and astragalin exhibit obvious hypnotic activity and have the potential to be commercialized as supplements for ameliorating central-nervous-system-related symptoms [10].

### 4.10. Antimicrobial Activity

The action of *E. ulmoides* on pathogenic microorganisms has been confirmed in vitro and in vivo, with broad-spectrum antibacterial and antifungal activity. Wu et al. compared and investigated the bioactivities of seven varieties of *E. ulmoides* on bacteria and fungi, and found that EUL extracts had more potent antibacterial activity than EUE extracts, and that all seven types of *E. ulmoides* showed broad-spectrum antibacterial and antifungal activity [168]. The EUL and EUE extracts of *E. ulmoides* exhibited antagonistic activity against *Escherichia coli*, *Pseudomonas aeruginosa*, *Salmonella paratyphi*, *Salmonella typhimurium*, *Bacillus subtilis*, *Staphylococcus aureus*, *Salmonella enteritidis*, *Pseudomonas aeruginosa*, and *Candida albicans*. Typically, the minimum inhibitory concentration (MIC) and minimum bactericidal concentration (MBC) values for all the tested bacterial strains ranged from 0.3125 to 10.00 mg/mL, and purple-leaf EUL, conventional EUL, and short-branch and dense EUL showed more potent activity against the bacteria than other *E. ulmoides* [1]. Additionally, *Bacillus velezensis* 157, isolated from EUE, exhibited antagonistic activity against a broad spectrum of pathogenic bacteria and fungi, for example, *Staphylococcus aureus*, *Escherichia coli*, *Salmonella typhimurium*, *Clostridium perfringens*, *Proteus hauseri*, *Aeromonas hydrophila*, *Streptococcus agalactiae*, *Botrytis cinerea* and *Fusarium oxysporum* [169]. Weaning piglets were fed with 0.01% *E. ulmoides* flavones (EUF) and appeared to show abundant gut microbiota and rising serum IgG levels [170]. Additionally, protocatechuic acid, extracted from *E. ulmoides*, increased the Firmicutes/Bacteroidetes ratio and reduced the relative abundance of *Prevotella* 9, *Prevotella* 2, *Holdemanella*, and the *Ruminococcus torques* group, and increased the relative abundance of *Roseburia* and *Desulfovibrio* in weaning piglets [171]. Except for EUF, bioactive substances in *E. ulmoides* also targeted the basic structure of bacteria. Fermentation is an essential process for many Chinese medicinal herbs. EUL vinegar was made under fermentation conditions, and exhibited an anti-*Bacillus subtilis* effect, relying on the destruction of bacterial cell walls and cell membranes to induce cell permeability [172]. Moreover, Hou et al. found that different wood vinegars produced from pyrolysis of *E. ulmoides* exerted antibacterial activity on *Enterobacter aerogenes*, *Escherichia coli*, *Staphylococcus aureus*, *Bacillus subtilis*, and *Bacillus cereus*; antifungal activity on *Penicillium*, *Rhizopus*, and *Aspergillus*; and anti-plant-pathogen activity on wheat root rot, buckwheat leaf spot, watermelon wilt, and eggplant wilt [173]. Furthermore, another study compared the effects of EUL ferment, EUL powder, and EUL extracts on gut microbiota in weaning piglets. It regarded EUL extract as a promising agent to balance gut microbiota that reduced the abundance of *Bacteroidetes* and enhanced the abundance of *Firmicutes* [174].

In a study on hepatitis B virus (HBV) in vitro, EUF showed anti-HBV activity in the dose range of 25–200 μg/mL, and significantly inhibited the replication of HBV-DNA and the secretion of HBeAg and HBsAg [175]. Huang et al. illustrated that the ethanol extracts of *E. ulmoides* had antagonistic activity against white spot syndrome virus replication in crayfish, *Procambarus clarkii*, with a death value of 84.12%, and in which geniposidic acid may be the main active constituent [176]. Moreover, one study suggested that extracts of *E. ulmoides* be used as an additive in animal feed to reduce the use of hormones and antibiotics during the breeding process [177].

## 5. Preclinical and Clinical Studies

There are only a few clinical studies on *E. ulmoides* or its active components. The relationship between obesity and the NEI axis has been described in the previous sections. An early clinical study on 27 abdominal obesity patients confirmed that EUL extract reduced body weight and fat. The patients were divided into two groups. The 15 patients in the experimental group were administrated with 1500 mg EUL extract per day, while the rest were administrated with placebo capsules. The EUL extract significantly reduced the accumulation of subcutaneous fat and visceral fat after 8 weeks, and decreased body weight and waist circumference were observed in the fourth and eighth weeks [178]. The mechanisms of primary infertility and recurrent spontaneous abortion have been reported to involve immune-mediated pathways, including elevated levels of NK cells [179]. Compared with progesterone, concomitant treatment with *E. ulmoides* granules significantly relieved bleeding, sore waists, and abdominal pain in 130 abortion threatened patients, as demonstrated by B-scan ultrasonography and vaginal symptoms [180]. Osteoarthritis is caused by immune–neuroendocrine dysregulation, affecting both systemic inflammatory and stress mediators and the function of innate immune cells [181]. A multicenter, randomized, double-blind, placebo-controlled clinical study is ongoing to evaluate the efficacy of *E. ulmoides* extract in mild osteoarthritis patients [182].

## 6. The Pharmacology Network of *Eucommia ulmoides* Oliver

The construction of a pharmacology network serves to clarify the potential molecular mechanism for the active components of *E. ulmoides*. We retrieved active ingredients of *E. ulmoides* from the literature and the TCMSP database (https://tcmsp-e.com/, accessed on 2 October 2021) [183] with criteria of oral bioavailability (OB) ≥ 30% and drug likeness (DL) ≥ 0.18. Ultimately, 33 compounds and 238 putative targets were screened (Figure 5A). According to the degree of compounds and targets, it has been demonstrated that PTGS2, CALM1, HSP90AA1, and NCOA2 are putative targets, while the important components are Quercetin (DZ30), Kaempferol (DZ31), (-)-Tabernemontanine (DZ14), and Epiquinidine (DZ8). The compound–target network was visualized using Cytoscape (vision 3.9.0) [184] (Figure 5A). In addition, the KEGG enrichment results illustrated that *E. ulmoides* is potentially involved in functional interactions with cancer, hepatitis, insulin resistance, the FoxO signaling pathway, non-alcoholic fatty liver disease, and the Toll-like receptor signaling pathway (Figure 5B). These results are consistent with our previous literature review, showing that *E. ulmoides* has a close connection with NEI-associated disease, especially cancer.

## 7. *Eucommia ulmoides* Oliver-Based Products

*E. ulmoides* has been used as a medicine in markets with various formulas. In *Chinese Pharmacopeia* (2020 version), *E. ulmoides* is used in 44 medical prescriptions, usually with *Gastrodia elata* Bl. (Tian-ma), *Angelica pubescens* Maxim. f. biserrata Shan et Yuan (Du-huo), and *Angelica sinensisn* (Oliv.) Diels (Dang-gui) (Table 2). Among these formulas, the majority (23 formulations) contain *E. ulmoides* processed with salt, exhibiting properties of tonifying the kidney and invigorating the blood, thus achieving relief of back or knee pain, regulating gynecological disorders, and preventing miscarriage (Figure 6A). In the Taiwan Chinese medicine benchmark, five prescriptions including *E. ulmoides* were recorded, namely Huan-Shao-Dan pills, San-Bi decoction, Du-Huo-Ji-Sheng decoction, Tiao-Jing Pills, and Bu-Yin decoction. On the China National Medical Product Administration’s (NMPA’s) website using “Duzhong” or “*E. ulmoides* Oliv.” As keywords, we found 59 types of drugs active in the market, in the form of tablets (18), capsules (17), particles (14), and others (Table 3).

Additionally, *E. ulmoides* has been formulated as a dietary supplement or nutraceutical. There are about 56 kinds of dietary supplements or nutraceuticals using *E. ulmoides* in China, for therapeutic supplementary effects on NEI-related disease. These effects include relieving hepatic injury, strengthening bones, boosting immunity, reducing blood pressure and blood fat, and alleviating physical fatigue with dosage forms such as capsules, teas, other beverages, and oral liquids (information from the China NMPA’s website (Figure 6B)). From the Dietary Supplement Label Database of the U.S. National Institute of Health, we found 73 registered dietary supplements in the form of capsules, powder, tablets, oral liquid, and sachets that contain *E. ulmoides*. In Japan, the main health products that include *E. ulmoides* are prepared as flavored beverages, used to assist the treatment of hypertension in a specific population (Figure 6C).

## 8. Conclusions

*E. ulmoides* is a plant with a wide range of bioactive compounds used for the treatment of a variety of diseases, especially NEI-related ones, in the form of extracts or enriched active ingredients. In this review, we established that *E. ulmoides* extracts and their compounds possess the ability to treat NEI-associated diseases, such as neurological disorders, hyperlipidemia, hypertension, obesity, and cancers. The *E. ulmoides* extracts and their compounds also have modulating effects on bone and hormones, in both in vivo and in vivo models, and are involved in multiple biological pathways. A functional enrichment analysis of the pharmacological effects of *E. ulmoides* was conducted through network pharmacology, and we found effects related to cancers and neurological and metabolic diseases, a finding similar to the results of the literature review. Therefore, *E. ulmoides* is a promising drug for the treatment of NEI-associated diseases.

However, the current study has some limitations. We have analyzed information on *E. ulmoides* products available in the Chinese market, and note that when used as a medicine, *E. ulmoides* is usually pre-treated in different ways for use as a treatment to promote liver and kidney function, strengthen muscles and bones, and improve immunity. While these biological effects do correlate with the NEI system, there is no specific disease which needs to be given priority in the future. Secondly, most of the bioactivity studies on *E. ulmoides* have been conducted on cells or animals, but relevant clinical studies were scarce and future clinical studies with *E. ulmoides* should be emphasized. Finally, as a plant with multiple compounds, its structure–activity relationship warrants further investigation in order to improve its development value and achieve efficiency gains.

## Figures and Tables

**Figure 1 molecules-27-03697-f001:**
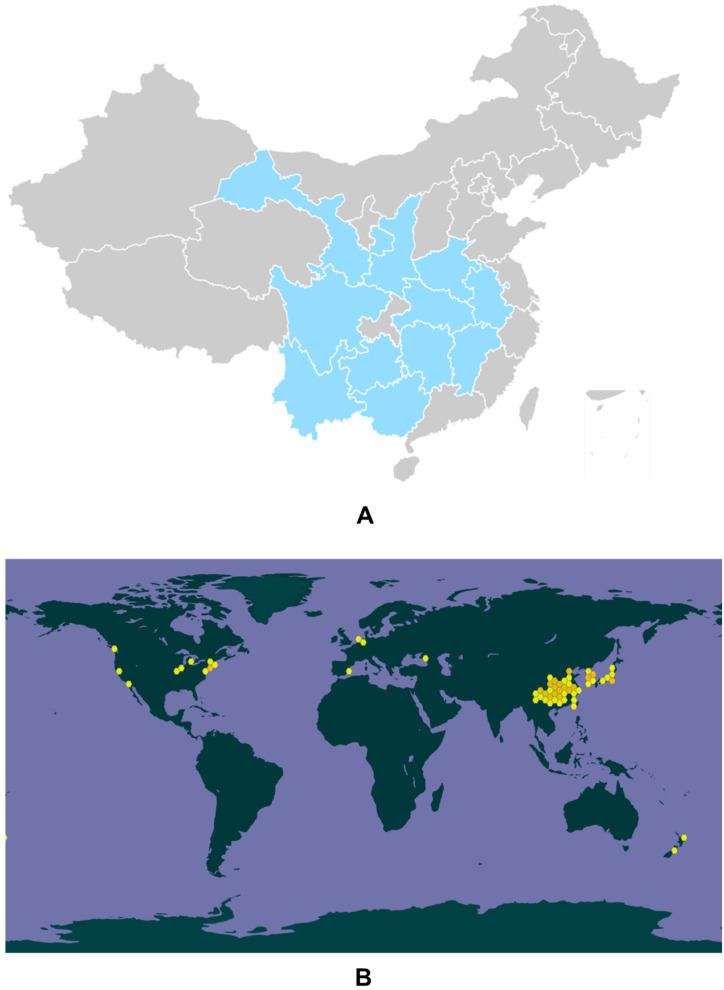
Genuine areas for *E. ulmoides* in China (**A**). The global biodiversity information of *E. ulmoides* (**B**).

**Figure 2 molecules-27-03697-f002:**
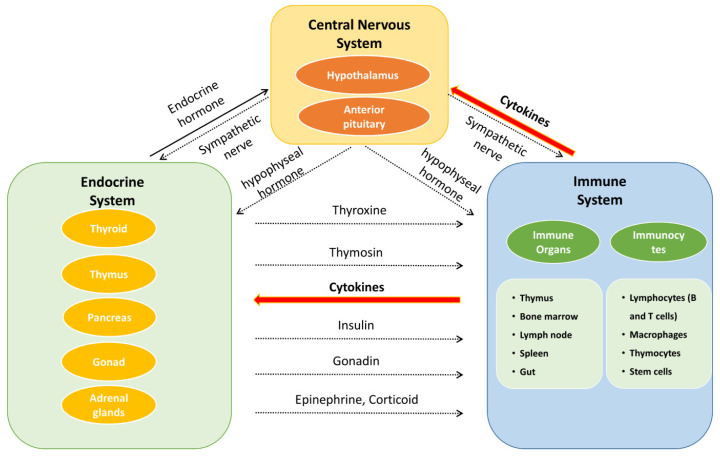
The interaction of neuroendocrine–immune system.

**Figure 3 molecules-27-03697-f003:**
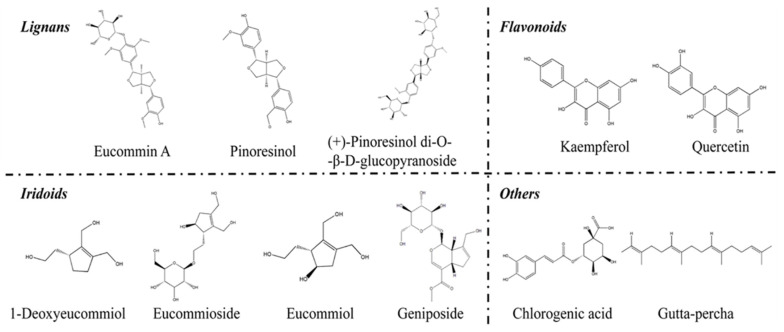
The main active chemical compounds in *E. ulmoides*.

**Figure 4 molecules-27-03697-f004:**
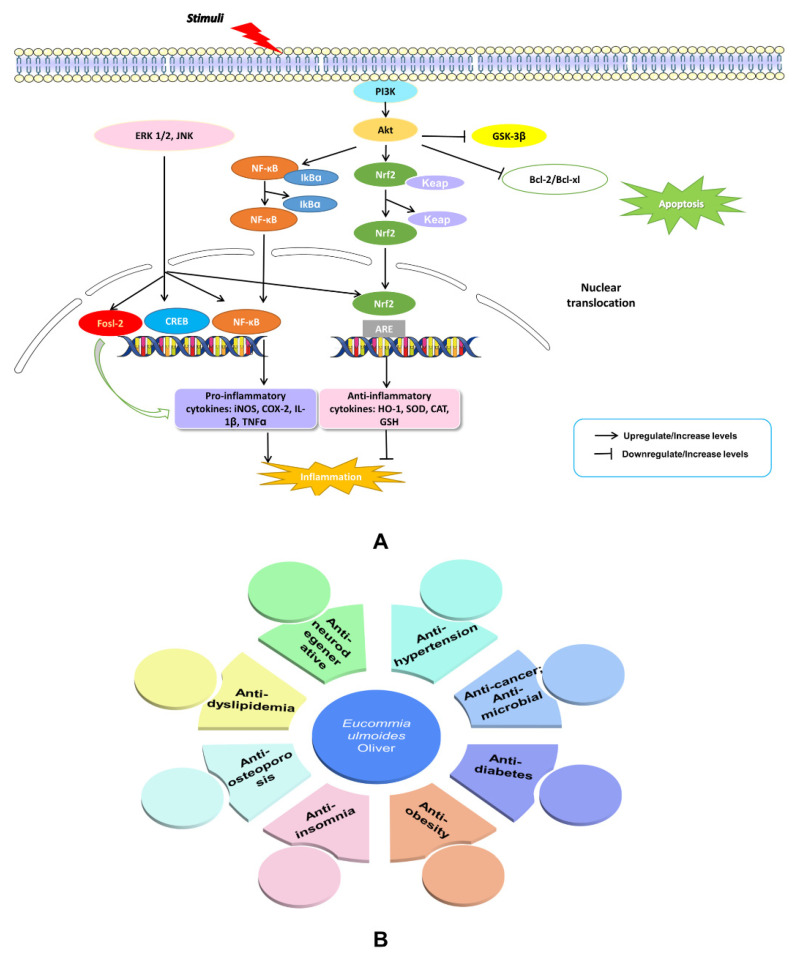
An illustration of NF-kB and PI3K-Akt signaling and their effects on AD and PD (**A**). Summary of published therapeutic properties of *E. ulmoides* (**B**).

**Figure 5 molecules-27-03697-f005:**
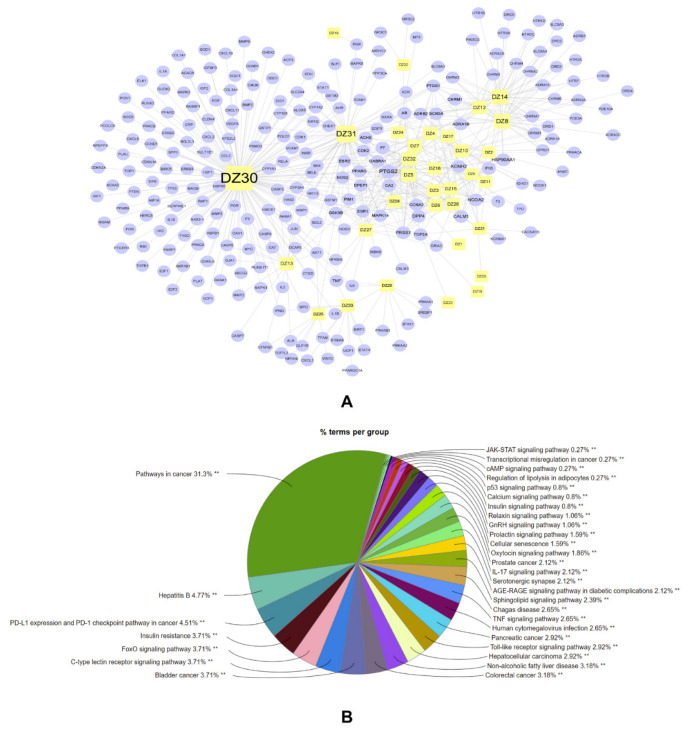
“Compound-target” network. Yellow circles represent compounds of *E. ulmoides*, purple circle represents targets (**A**). The proportion of enriched KEGG pathway of target (**B**). ** *p* < 0.01.

**Figure 6 molecules-27-03697-f006:**
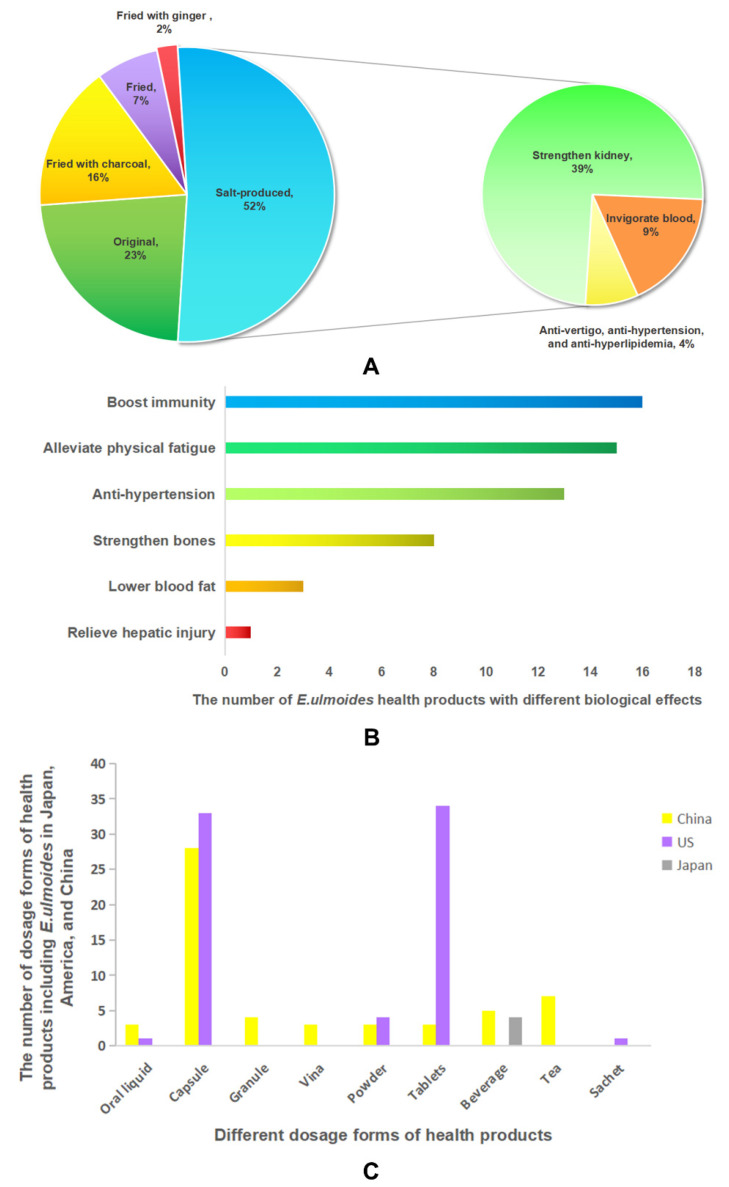
The proportion of *E. ulmoides* formulas: different processing approaches for *E. ulmoides* in 44 formulations (Large circle) and subplot for treatment features of salted *E. ulmoides* (Small circle) (**A**). The functions of health products including *E. ulmoides* in China (**B**). The number of dosage forms of health products including *E. ulmoides* in Japan, America, and China (**C**).

**Table 1 molecules-27-03697-t001:** Summary of pharmacological effects for *Eucommia ulmoides* Oliver.

Disease	Compound	Model		Dosage	Effect	Mechanism	Ref.
		In Vitro	In Vivo				
Cancer	Chlorogenic acid	AGS cells		0–2 mg/mL	Cytotoxicity		[51]
	Total flavonoids	GBMs cells lines U251, U87, HS683 and A172 and human normal cell HA	H_22_ tumor-bearing mice	50–200 mg/kg	Inhibit tumor growth Radiosensitization Induce apoptosis	Increase Bax expression and decrease in Bcl-2 expression; Decrease the ratio of Bcl-2/Bax and downregulate the expression of HIF-1α, MMP-2 as well as Wee1.	[52,53]
	Eucommicin A	iCSCL-10A-1, iCSCL-10A-2, MCF7, MDA-MB231 cells		0–100 μM	Cytotoxicity, suppressed tumor sphere formation		[54]
	Pentacyclic triterpenoids (betulinic acid, lupeol, and 3-O-laurylbetulinic acid)	Hela, MDA-MB-231, and T47D cells		3–80 μM	Inhibit tumor cell growth and induce apoptosis	Induce mitochondrial fragmentation and suppress lysosome production in Hela cells.	[55]
	Chlorogenic acid	HCT-116, LOVO		600–1600 µg/mL	Inhibit proliferation and promote apoptosis		[56]
	*Eucommia ulmoides* Oliver leaf (EUL) extract	A549, SNU-C4, HeLa,		25–200 µg/mL	Inhibit proliferation		[57]
	Total Polysaccharides	LLC, KMB-17		0.5–8.0 µg/mL	Induce apoptosis and inhibit proliferation	Activate Caspase-3 pathway.	[58]
	*E. ulmoides* extract	HCT116		500–800 mg/L	Cytotoxicity		[59]
	EUL extract and chlorogenic acid	HCT116, LOVO		1600 µg/mL	Inhibit invasion and migration		[60]
Alzheimer’s disease (AD) and Parkinson’s disease (PD)	*Eucommia ulmoides* Oliver bark (EUE) extract	Lipopolysaccharide (LPS)-stimulated BV-2 microglia 6-hydroxydopamine (6-OHDA)-induced SH-SY5Y cells		2.5–100 μg/mL	Anti-inflammatory Anti-oxidative stress	Inhibit phosphorylation of MAPKs, PI3K/Akt, and GSK-3β, suppress NF-κB activation and induce Nrf2-dependent HO-1 activation; Inhibit reactive oxygen species (ROS) production, mitochondrial dysfunction, and phosphorylation of JNK, PI3K/Akt and GSK-3β, thereby blocking NF-κB nuclear translocation.	[61,62]
	EUE extract	H_2_O_2_ -induced SH-SY5Y cells	Scopolamine-induced ICR mice	5–20 μg/mL, 5–20 mg/kg	Anti-cytotoxicity Enhance cholinergic signaling	Inhibit cytotoxicity, reduce ROS accumulation, DNA condensation, MMP stabilization, regulate Bcl-2 family proteins, inhibit MAPKs and PI3K/Akt phosphorylation; Decrease the activity of AChE and TBARS, protect BDNF and activate CREB expression.	[63,64]
	EUE extract		MPTP-induced male C57BL/6J mice	2.5–10 g/kg, 150–600 mg/kg	Anti-neuroinflammationAnti-PD	Downregulate expression of p38, JNK, and Fosl2, reduce pro-inflammatory factors; Antagonize loss of striatal neurotransmitters and alleviate associated ambulatory motor abnormalities.	[65,66]
	Betulin, wogonin, oroxylin A, geniposidic, aucubin	MPP+-induced SH-SY5Y cells		10 μM	Anti-PD	Ameliorate the ubiquitin-proteasome system.	[66]
	Geniposidic acid (GPA)		APP/PS1 mice and C57BL/6J mice	25, 75 mg/kg	Anti-neuroinflammatory	Inhibit the activation of astrocytes and microglia, down-regulate the expression of pro-inflammatory cytokines and iNOS, upregulate the expression of anti-inflammatory cytokines and Arg-1, and block the TLR4/2-MyD88 signaling pathway by reducing the expression of HMGB-1.	[67]
	Macranthoin G	Hydrogen peroxide (H_2_O_2_)-induced PC12 cells		6.25–50 μM	Anti-oxidative stress-mediated cellular injury Anti-PD and anti-AD	Decrease MDA production and ROS levels, increase MMP, restore CAT, GSH-Px and SOD activity, and inhibit NF-κB pathway and activation of IκBα, p38 and ERK.	[68]
Dsylipidemia	EUL extract		High-fat diet (HFD)-induced male Sprague-Dawley	200 mg/kg	Hepatoprotective	Inhibit ER stress, enhance lysosomal function, and increase autophagic flux associated with inhibition of the mTOR-ER stress pathway.	[69]
	EUE extract, aucubin and geniposide	Palmitate-induced HepG2 cells HFD-induced female Sprague-Dawley rats		100 μg/mL extracts, 10 μg/mL aucubin or geniposide	Anti-hepatic dyslipidemia	Inhibit ER stress by increasing V-ATPase activity, reduce hepatic lipid accumulation through secretion of apolipoprotein B and associated triglycerides and cholesterol; Enhance lysosomal activity and to regulate ER stress.	[70]
	EUE extract		CCl_4_-induced Sprague-Dawley rats	0.25–1 g/kg	Anti-hepatic dyslipidemia	Increase lysosomal enzyme activity, reduce ER stress by improving Apo B secretion, then inhibit ROS accumulation.	[71]
	EUE extract, aucubin, geniposide	BAX-induced HepG2 cells;	HFD-induced female Sprague-Dawley	100 μg/mL extracts, 10 μg/mL aucubin or geniposide; 0.25–1 g/kg;	Anti-hepatic dyslipidemia	Inhibit cell death through enhancement of lysosome activity; Enhance lysosomal activity to the regulate lysosomal BAX activation and cell death.	[72]
	CGA enriched-EUL extract	HepG2 cells		10–80 mg/L; 0.3–600 μM;	Lipid-lowering	Activate AMPK and inhibit SREBP2 and HMGCR to reduce TC synthesis and TG levels, increase ABCA1 and CYP7A1, and enhance TC excretion and bile acid transport, synthesis and excretion.	[73]
	Total flavonoid		HFD-induced male Wistar rats	10–90 mg/kg/day	Anti-hyperlipidemia	Lower serum cholesterol, triglyceride, lipoprotein, apolipoprotein, and density lipoprotein cholesterol levels, increase HDL cholesterol and apolipoprotein A.	[74]
Osteoporosis	Total lignans	Primary cultures of rat osteoblasts	Ovariectomy rat model	20, 40, or 80 mg/kg/day; 300 μg/mL	Anti-osteoporosis, prevent OVX-induced decrease of bone mass and deterioration of trabecular microarchitecture	Induce primary osteoblastic cell proliferation and differentiation; Increase osteoprotegrin expression and decrease NF-κB ligand expression.	[75]
	EUE extract		Adolescent female rats	30, 100 mg/kg	Increase longitudinal bone growth rate and enhance osteoblastogenesis	Promote chondrogenesis in the growth plate and increase BMP-2 and IGF-1.	[76]
	5-(hydroxymethyl)-2-furaldehyde (5-HMF)	Rat bone mesenchymal stem cells (bMSCs)		0.05, 0.10, and 0.20 mg/mL	Anti-osteoporosis; inhibit adipogenesis and enhance osteoblastogenesis	Increase ALP, COL1alpha1 (7 days only), OCN and OPN expression, decrease PPARgamma, FABP4, C/EBPalpha and LPL expression.	[77]
	Pinoresinol 4′-O-β-d-glucopyranoside, pinoresinol di-O-β-d-glucopyranoside, aucubin, wogonin, baicalein, and α-O-β-d-glucopyranosyl-4,2′,4′-trihydroxydihydrochalcone	MCF-7 cells; MDA-MB-231 cells; Hela cells		10^−6^ M, 10^−5^ M, and 10^−4^ M	Prevent estrogen deficiency-induced osteoporosis	Activate ER-dependent transcription of estrogen target genes; Exhibit significant difference in ER subtype (α vs. β) selectivity; Proliferation effect on breast cancer cells mediated by the genomic action of Erα. Stimulation of endogenous estrogen-responsive genes (pS2).	[78]
	EUL extracts	Rat osteoblastic MC3T3-E1 cells		6.25, 12.5, 25, 50, and 100 µg/mL	Anti-osteoporosis, restrain cell oxidative damage and increase cell survival rate in a dose-dependent manner	Decrease the expression of caspases 3, 6, 7, and 9.	[79]
Insomnia	Astragalin; Eucommiol		KM mice	5, 10 and 20 mg/kg; 50, 100, and 200 mg/kg		Reduce spontaneous activity, increase sleep ratio, shorten sleep latency and lengthen sleep time; Reduce the convulsion rate and prolong convulsion latency.	[80,81]
Hypertension	Total flavonoid	Human glioblastoma cells (U251, U87, HS683 and A172)		0.5–32 μg/mL	Enhance the radiotherapy effect, decrease the cell viability, inhibit migration and invasion,	HIF-α/MMP-2 pathway and intrinsic apoptosis pathway.	[52]
	Male flower extract		Male spontaneously hypertensive rats, Sprague Dawley rats	0.05, 0.10, 0.20 g/mL	Reduce blood pressure, promote the expression of ACE2	Activate the ACE2-Ang-(1–7)-Mas signaling pathways.	[82]
	EUL extract		Wistar-Kyoto rats	5% (*w*/*w*, extract/high-fat diet)	Reduce blood pressure, prevent aortic media hypertrophy		[11]
Diabetes mellitus	EUE extract		Streptozotocin (STZ)-induced diabetic rat model	1.4 g/kg	Reduce the level of plasma glucose	Prohibit the reduction of superoxide dismutase (SOD) activity; Suppress the elevation of malondialdehyde (MDA).	[83]
	EUL extract and EUL powder		HFD-induced male SD rats	3%, 9% EUL 3%, 9% EGLP	Improve insulin resistance and decrease plasma glucose level, reduce the production of ATP and the level of triacylglyceride, and regulate fatty acid oxidation	Enhance the use of circulating blood glucose in skeletal muscles.	[84]
	Asperuloside		HFD-induced male SD rats	0.03, 0.1, 0.3 ASP; 5% ELE	Reduce body weight, visceral fat, food intake, and circulating levels of glucose, insulin, triacylglyceride and nonesterified fatty acid	Increase mRNA levels of Cs, Idh3α, Ogdh, Sdha, Comp I, Comp IV, and Comp V in skeletal muscles; Reduce ATP production in WAT; Increase mRNA level of FA transport protein, Cpt1α and Acadvl, suppress Fas mRNA, and activate FA β-oxidation.	[85]
	5% chlorogenic acids contained in ELE	HepG2 cells		200, 400, 500 μg/mL	Promote glucose uptake	Inhibit glucose-6-phosphate displacement enzyme and α-glucosidase.	[86]
	*E. ulmoides*		STZ induced- type 1-like DM rats	1 g/kg/day oral administration	Decrease the level of blood urea nitrogen and creatinine, improve renal fibrosis, without influencing blood glucose level	Inhibit TGF-β/Smad signaling pathway and suppress expression of TGF-β/connective tissue growth factor.	[87]
	EUE extract		STZ-induced mice	200 mg/kg oral administration	Inhibit production of advanced glycation end products (AGEs) and AGEs receptors	Increase the Glo1 expression and activity; Elevate Nrf2 protein expression and reduce RAGE expression.	[88]
	Isoquercetin, 6″-O-acetyl-astragalin, kaempferol, quercetin, rutin, kaempferol 3-O-rutinoside, astragalin	Ribose-gelatin		0.01, 0.1, 1, 10, 100 μg/mL	Inhibit the formation of AGEs	Block the formation of CML and CMA.	[89]
	Lignans	RF/6A cells	STZ-induced male C57BL/6 mice	25, 50, 75, and 100 μg/mL	Protect endothelial function from AGEs injury and oxidative stress	Regulate Nrf2/HO-1 signaling pathway.	[16]
	Lignans	RMCs (HBZY-1 cells)		20, 40, and 80 mg/L	Inhibit the proliferation of mesangial cells	Reduce the mRNA expression of Col I, Col III, Col IV, and fibronectin; Reverse the elevation of aldose reductase.	[90]
Obesity	Asperuloside		Male C57BL/6J mice	0.25% (*w*/*w*)	Reduce liver, epididymal, and mesenteric white adipose tissue, decrease serum triglyceride level	Increase *Akkermansia*, *Parabacteroides*, *Bacteroides*, *Sutterella*, *Anaerostipes*, *Roseburia*, and *Coprobacillus* abundance Change metabolic level of cecum, Inhibit GLP-1; Reduce the level of tumor necrosis factor alpha (TNFα), monocyte chemoattractant protein 1 (MCP1), and collagen type 1 alpha1 (Col1a1) Increase lipoprotein lipase (Lpl) and carnitine palmitoyl transferase 1 (Cpt1).	[91]
	EUL extract Asperuloside		HFD-induced male SD rats	0.03, 0.1, 0.3 ASP; 5% ELE	ASP reduce the body weight, visceral fat, food take, triacylglyceride and nonestesterified fatty acid	Diminish dehydrogenase;Increase Glut4, succinyl CoA synthase; Increase mRNA levels of Cs, Idh3α, Ogdh, Sdha, Comp I, Comp IV and Comp V in skeletal muscles; Increase uncoupling protein 1 in brown adipose tissue mRNA;Reduce ATP production in WAT; Increase mRNA level of FA transport protein, Cpt1α and Acadvl, suppress Fas mRNA, and activate FA β-oxidation.	[85]
	Quercetin				Reduce fat accumulation in liver	Decrease the level of plasma lipid.	[92]
	ELE, ELE aroma		Male SD rats	5% ELE	Promote metabolism of lipid	Elevate the level of Cpt2, Acad, complex II and V mRNA in liver; Increase expression of brain-derived neurotrophic factor, protein kinase, and phospholipase Cγ in hypothalamus.	[93]
	ELE extract		Male Wistar-Kyoto rats	5% ELE	Reduce the body weight gain, visceral and perirenal fat		[11]
	CGA-enriched extract from EUE	HepG2 cells		10, 20, 25, 40, 60, and 80 mg/L	Reduce the lipid in HepG2 cells	Elevate the expression of ABCA1, CYP7A1, and AMPKα2; Reduce the level of SREBP2 and inhibit mRNA and expression of HMGCR.	[73]

**Table 2 molecules-27-03697-t002:** The prescriptions including *Eucommia ulmoides* Oliver in China Pharmacopeia.

No.	Formulas	Dosage Form	Pharmacological Effects
1	San-Bao capsule	Capsule	Strengthen kidney and fill essence, nourish heart, and calm mind. It is used for deficiency of kidney essence and deficiency of heart and blood, resulting in weakness of the waist and legs, impotence and spermatorrhea, dizziness, tinnitus and deafness, palpitation, insomnia, and loss of appetite.
2	Tian-Zi-Hong-Nv-Jin capsule	Capsule	Benefit qi and nourish blood, tonify the kidney and warm the uterus, used for qi and blood deficiency, kidney deficiency and coldness in the uterus, irregular menstruation, cold pain in the waist and knees, cold and infertility in the uterus.
3	Quan-Du-Zhong capsule	Capsule	Tonify the liver and kidney, strengthen the muscles and bones, and lower blood pressure. Used for kidney deficiency and lumbar pain, weakness of the waist and knees; hypertension.
4	Gui-Ling-Ji	Capsule	Strengthen the body and nourish the brain, strengthen the kidneys and tonify the qi, and increase appetite. It is used for kidney deficiency and Yang weakness, memory loss, night dreams with semen overflow, waist thinness and leg weakness, and cough with qi deficiency.
5	Zhi-Mai-Kang capsule	Capsule	It is used to reduce food intake, lower lipids, promote blood circulation, and benefit qi and blood. Used for arteriosclerosis and hyperlipidemia caused by internal stagnation and deficiency of qi and blood.
6	Ha-Ha-Bu-Shen capsule	Capsule	Strengthen the Yang and benefit the kidney, fill the essence and replenish the blood. For physical weakness, deficiency of vital energy, frequent urination.
7	Qiang-Li-Ding-Xuan capsule	Capsule	Lower blood pressure, lipids, and dizziness. For hypertension, arteriosclerosis, hyperlipidemia, headache, dizziness, dizziness, tinnitus, and insomnia caused by the above diseases.
8	Tian-Ma-Gou-Teng granule	Granule	Calm the liver and quench wind; clear heat and calm the mind. For headache, dizziness, tinnitus, blurred vision, tremor, insomnia caused by hyperactivity of liver Yang; hypertension.
9	Tian-Zhi granule	Granule	Calm the liver and submerge the Yang, tonify the liver and kidney, educate and calm the mind. For dizziness, headache, insomnia, irritability, dryness of the mouth and throat, weakness of the waist and knees, loss of intelligence, slow thinking, poor orientation, and mild to moderate vascular dementia.
10	Yun-Kang granule	Granule	Strengthen the spleen and kidney, nourish blood, and calm the fetus. It is used for kidney deficiency and qi and blood weakness type of pre-eclampsia and habitual miscarriage.
11	Fu-Bao granule	Granule	Benefit the kidney and blood, regulate qi and relieve pain. It is used in treating weakness of the kidney and stasis in the lower back and legs, distension and pain in the abdomen, leucorrhea, and menstrual leakage; chronic pelvic inflammatory disease and adnexitis.
12	Ji-Sheng-Zhui-Feng alcohol	Medicinal wine	Tonify the liver and kidney, dispel wind dampness, and relieve paralysis and pain. It is used for deficiency of liver and kidney, wind cold and dampness paralysis, cold pain in the waist and knees, unfavorable flexion, and extension; rheumatic arthritis, lumbar muscle strain, late stage of bruises and injuries.
13	Du-Huo-Ji-Sheng mixture	Mixture	Nourish blood, relax tendons, dispel wind and dampness, and nourish the liver and kidney. It is used for paralysis caused by wind, cold and dampness, deficiency of liver and kidney, and deficiency of qi and blood, which results in cold pain in the waist and knees and unfavorable flexion and extension.
14	Tian-Jing-Bu-Shen ointment	Ointment	Warm the kidney and help Yang, tonify essence and blood. Used for deficiency of kidney Yang and deficiency of essence and blood, resulting in weakness of the waist and knees, depression, fear of cold, impotence and spermatorrhea.
15	Yun-Kang oral liquid	Oral liquid	Strengthen the spleen and kidney, nourish blood and calm the fetus. Used for kidney deficiency, qi and blood weakness, pre-term abortion and habitual abortion.
16	Hen-Gu-Gu-Shang-Yu-He	Oral liquid	Promote healing of fractures by invigorating the blood, tonify the liver and kidneys, connect bones and tendons, relieve swelling and pain. It is used for fresh and old fractures, femoral head necrosis, osteoarthrosis, lumbar intervertebral disc exacerbation.
17	Qian-Jin-Zhi-Dai pill	Pill	Strengthen the spleen and tonifying the kidneys, regulate menstruation and stop banding. It is used for menstrual disorders caused by deficiency of the spleen and kidneys, and for the disease of hypermenorrhea, which is characterized by irregular menstrual flow, large amount of menstruation or dripping without lumpiness, or a large amount of hypermenorrhea with white and thin color, fatigue, and weakness of the waist and knees.
18	Tian-Ma pill	Pill	Dispel wind and dampness, relieve pain, tonifying the liver and kidney. It is used for paralysis caused by wind dampness and stagnation and liver–kidney deficiency, which is characterized by constriction of limbs, numbness of hands and feet, and pain in the waist and legs.
19	Dang-Gui-Bu-Xue pill	Pill	Benefit qi and nourish blood to regulate menstruation. For menstrual disorders caused by deficiency of qi and blood, such as early menstruation, low or high menstrual blood volume, prolonged menstrual period, weakness of limbs.
20	Quan-Lu pill	Pill	Tonify the kidney and fill the essence, strengthen the spleen, and benefit the qi. Used for the elderly with weakness of the waist and knees, fatigue, cold in the extremities, and frequent urination due to deficiency of both spleen and kidney.
21	Fu-Ke-Yang-Kun pill	Pill	Diversify the liver and qi, nourish the blood, and invigorate the blood. Used for irregular menstruation, amenorrhea, dysmenorrhea, and menstrual headache caused by blood deficiency and liver depression.
22	Fu-Ke-Yang-Rong pill	Pill	Nourish qi and blood, relieve liver and depression, eliminate blood stasis, and regulate menstruation. Used for deficiency of qi and blood, liver depression, menstrual disorders, dizziness, blood leakage and blood collapse, anemia, and infertility.
23	Shen-Jin-Huo-Luo pill	Pill	Relax tendons and activates collaterals, dispel wind and dampness, warm menstruation and relieve pain. It is used for paralysis caused by wind, cold and dampness, blocking the arteries and veins, and is associated with cold and painful joints of the limbs, unfavorable flexion and extension, numbness of the hands and feet, and paralysis of the body.
24	Miao-Ji pill	Pill	Tonify the liver and kidneys, dispel dampness and promote circulation, activate blood circulation and relieving pain. It is used for paralysis caused by deficiency of liver and kidney, wind-dampness, and stagnation, and is associated with pain in the bones, soreness and weakness of the waist and knees, and numbness and constriction of the limbs.
25	Qing-E-pill	Pill	Tonify the kidney and strengthen the waist. For kidney deficiency and lumbar pain, unfavorable starting and sitting, and weakness of the knees.
26	Shen-Rong-Bao-Tai pill	Pill	Nourish the liver and kidney, tonify the blood and calm the fetus. Used for deficiency of liver and kidney, deficiency of blood, physical weakness, pain in the waist and knees, abdominal cramps, bleeding in pregnancy and fetal disturbance.
27	Du-Huo-Ji-Sheng pill	Pill	Nourish blood, relax tendons, dispel wind and dampness, and nourish the liver and kidney. It is used for paralysis caused by wind, cold and dampness, deficiency of liver and kidney, and deficiency of qi and blood, which results in cold pain in the waist and knees and unfavorable flexion and extension.
28	Jian-Nao-Bu-Shen tablet	Pill	Strengthen the brain, tonify the kidneys, benefit the qi and strengthen the spleen, calm the mind, and fix the will. It is used for forgetfulness, insomnia, dizziness, tinnitus, palpitations, weakness of the waist and knees, seminal emission due to deficiency of the spleen and kidneys; neurasthenia and sexual dysfunction with the above symptoms.
29	Pei-Kun pill	Pill	Tonify qi and blood, nourish the liver and kidney. Used for women with blood deficiency, poor digestion, irregular menstruation, abdominal cold pain, weakened qi and blood, and prolonged infertility.
30	Hu-Po-Huan-Qing pill	Pill	Tonify the liver and kidney, clear heat and brighten the eyes. For internal and external cataracts, dilated pupils, diminished vision, night blindness, blurred vision, shyness of the eyes, and tears in the wind due to deficiency of the liver and kidneys and inflammation of deficiency fire.
31	Suo-Yang-Gu-Jing pill	Pill	Warm the kidney and consolidate the essence. For soreness of the waist and knees, dizziness, and tinnitus, spermatorrhea and premature ejaculation due to deficiency of kidney Yang.
32	Shu-Jin pill	Pill	Dispel wind and dampness, relax tendons, and activate blood. Used for wind-cold dampness paralysis, numbness of limbs, tendon pain and difficulty walking.
33	Qiang-Li-Tian-Ma-Du-Zhong pill	Pill	Disperse wind and invigorate blood, sooth tendons and relieve pain. It is used for painful strokes, numbness of the limbs, inconvenience in walking, soreness of the back and legs, headache, and dizziness.
34	Shu-Feng-Ding-Tong pill	Pill	Dispel wind and disperse cold, activate blood circulation, and relieve pain. It is used for paralysis caused by wind, cold, and dampness, and blood stasis.
35	Yao-Tong pill	Pill	Tonify the kidney and invigorate blood, strengthen tendons, and relieve pain. For low back pain and lumbar muscle strain caused by deficiency of kidney Yang and stasis of blood.
36	Tian-Ma-Qu-Feng-Bu tablet	Tablet	Warm the kidney and nourish the liver, expel wind and relieve pain. It is used for paralysis caused by deficiency of liver and kidney and wind dampness entering the ligaments, which is characterized by dizziness and tinnitus, joint pain, weakness of the waist and knees, fear of cold, cold limbs, and numbness of the hands and feet.
37	Shen-Yan-Kang-Fu tablet	Tablet	Benefit qi and nourish yin, strengthen the spleen and tonify the kidneys, and clear residual toxins. It is used for edema caused by deficiency of qi and yin, deficiency of spleen and kidney, and water and dampness.
38	Shen-Rong-Gu-Ben tablet	Tablet	Tonify qi and nourish blood. Used for tiredness of the limbs, dullness of the face, tinnitus, and dizziness due to deficiency of qi and blood.
39	Yao-Tong tablet	Tablet	Tonify the kidney and invigorate blood, strengthen tendons, and relieve pain. For low back pain and lumbar muscle strain caused by deficiency of kidney Yang and blood stasis.
40	Er-Shi-Qi-Wei-Ding-Kun pill	Pill	Tonify qi and nourish blood, relieve depression and regulate menstruation, for deficiency of qi and blood, weakness of body, irregular menstruation, menstrual disorders, abdominal pain during menstruation, menorrhagia, lumbago, and weakness of legs.
41	Wu-Bi-Shan-Yao pill	Pill	Strengthen the spleen and tonify the kidney. It is used for deficiency of spleen and kidneys, less food and muscle thinning, soreness and weakness of the waist and knees, dizziness, and tinnitus.
42	Danlu Tongdu tablet	Tablet	Promote blood circulation and benefit the kidney and the ligament. For intermittent claudication, lumbar and leg pain, restricted movement, soreness and pain in the lower extremities, and dark tongue or petechiae in lumbar spinal stenosis (e.g., thickening of the ligamentum flavum, degenerative changes of the vertebral body, and old disc protrusion) caused by stasis obstructing the directing vessel.
43	You-Gui pill	Pill	Warm kidney Yang, fill essence and stop spermatorrhea. It is used for deficiency of kidney Yang, failure of the vital gate fire, coldness of the waist and knees, mental weakness, coldness, and fear of cold, impotence and spermatorrhea, tantalizing thin stools, frequent and clear urination.
44	Qiang-Shen tablet	Tablet	Tonify the kidney and fill the essence, benefit the qi and strengthen Yang. It is used for edema, lumbago, spermatorrhea, impotence, premature ejaculation, frequent nocturia caused by deficiency of both yin and Yang; chronic nephritis and long-standing pyelonephritis with the above symptoms.

**Table 3 molecules-27-03697-t003:** The domestic drugs including *Eucommia ulmoides* Oliver in China.

No.	Domestic Drugs	Number	Dosage Form	Pharmacological Effects
1	Quan-Du-Zhong capsule	1	Capsule	Lower blood pressure, tonify the liver and kidney, and strengthen the muscles and bones. Used for hypertension, kidney deficiency and lumbar pain, weakness of the waist and knees.
2	Du-Zhong-Bu-Tian-Su capsule	1	Capsule	Warm the kidney and nourish the heart, strengthen the waist, and calm the mind. Used for soreness and weakness of the lumbar spine, excessive urination at night, neurasthenia.
3	Du-Zhong-Ping-Ya capsule	1	Capsule	Lower blood pressure and strengthen tendons and bones. Indicated for high blood pressure, dizziness, soreness of the waist and knees, and impotence of the tendons and bones.
4	Du-Zhong-Zhuang-Gu capsule	2	Capsule	Benefit qi, strengthen the spleen, nourish the liver and waist, activate blood circulation, strengthen the muscles and bones, dispel wind, and dampness. Used for rheumatic paralysis, weakness of tendons and bones, unfavorable flexion and extension, difficult gait, pain in the waist and knees, fearing cold and preferring warmth.
5	Fu-Fang-Du-Zhong capsule	2	Capsule	For hypertension due to kidney deficiency and liver exuberance.
6	Qiang-Li-Tian-Ma-Du-Zhong capsule	9	Capsule	It is used for the painful restraint of tendons and veins caused by stroke, numbness of limbs, inconvenience in walking, soreness of back and legs, headache, and dizziness.
7	Fu-Fang-Du-Zhong-Zhuang-Yao capsule	1	Capsule	It is used for lumbago and knee weakness caused by kidney deficiency, aggravated by exertion, lack of mobility and lack of warmth in the hands and feet.
8	Du-Zhong granule	13	Granule	Tonify the liver and kidney, strengthen the muscles and bones, calm the fetus, and lower blood pressure. Used for kidney deficiency and lumbago, weakness of the waist and knees, fetal restlessness, pre-term abortion, hypertension.
9	Fu-Fang-Du-Zhong-Jian-Gu granule	1	Granule	It is used for swelling, pain and dysfunction caused by osteoarthritis of the knee joint.
10	Du-Zhong medical wine	1	Medical wine	Warm the liver and kidney, tonify the qi and blood, strengthen the tendons and bones, dispel wind and dampness. For deficiency of liver and kidney, impotence of tendons and bones, wind cold and dampness paralysis.
11	Fu-Fang-Du-Zhong-Qiang-Yao medical wine	1	Medical wine	For kidney Yang deficiency caused by lumbar pain, unfavorable rotation, soreness and weakness of the waist and knees, tiredness, and weakness auxiliary treatment.
12	Du-Zhong-Bu-Yao mixture	1	Mixture	Tonify the liver and kidney, benefit the vital energy and blood, and strengthen the waist and knees. Used for pain in the waist and legs, fatigue and weakness, mental weakness, and frequent urination.
13	Fu-Fang-Du-Zhong-Fu-Zheng mixture	1	Mixture	For soreness and weakness of the waist and knees, tiredness and fatigue due to deficiency of the spleen and kidneys, loss of appetite, shortness of breath and fatigue.
14	Du-Zhong-Zhuang-Gu pill	1	Pill	Benefit qi, strengthen the spleen, nourish the liver and waist, activate blood circulation, strengthen tendons and bones, dispel wind and remove dampness. For the treatment of rheumatic paralysis, weakness of tendons and bones, unfavorable flexion and extension, difficult gait, pain in the waist and knees, fearing cold and preferring warmth.
15	Fu-Fang-Du-Zhong pill	1	Pill	Tonify the kidney, calm the liver and clear heat. Used for hypertension due to kidney deficiency and liver hyperactivity.
16	Shen-Qi-Du-Zhong pill	1	Pill	Benefit qi and tonify the kidney. For tiredness and fatigue, soreness and weakness of the waist and knees, forgetfulness, and insomnia.
17	Du-Zhong-Bu-Tian-Su pill	1	Pill	Warm the kidney and nourish the heart, strengthen the waist, and calm the mind. Used for soreness and weakness of the lumbar spine, excessive urination at night, neurasthenia
18	Qiang-Li-Tian-Ma-Du-Zhong pill	1	Pill	It is used for the painful restraint of tendons and veins caused by stroke, numbness of limbs, inconvenience in walking, soreness of back and legs, headache, and dizziness.
19	Du-Zhong-Ping-Ya tablet	10	Tablet	Tonify the liver and kidney, strengthen tendons and bones. For dizziness and dizziness caused by deficiency of liver and kidney, soreness of the waist and knees, impotence of tendons and bones, hypertension.
20	Du-Zhong-Jiang-Ya tablet	3	Tablet	Tonify the kidney, calm the liver and clear heat. For hypertension due to kidney deficiency and liver exuberance.
21	Fu-Fang-Du-Zhong tablet	3	Tablet	Tonify the kidney, calm the liver and clear heat. For hypertension due to kidney deficiency and liver hyperactivity.
22	Du-Zhong-Bu-Tian-Su tablet	1	Tablet	Warm the kidney and nourish the heart, strengthen the waist, and calm the mind. Used for soreness and weakness of the lumbar spine, excessive urination at night, neurasthenia.
23	Du-Zhong-Ping-Ya dispersion tablet	1	Tablet	Lower blood pressure and strengthen tendons and bones. It is suitable for high blood pressure, dizziness, soreness of the waist and knees, and impotence of tendons and bones.
24	Du-Zhong-Shuang-Jiang-Dai tea	1	Tea	It has the effect of lowering blood pressure and lipid. Used for hypertension and hyperlipidemia.

## Data Availability

Not applicable.

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
