# Peer review of "Neuroendocrine–Immune Regulatory Network of Eucommia ulmoides Oliver"

_molecules, 2022, doi:10.3390/molecules27123697_

Round 1

Reviewer 1 Report

The manuscript entitled Neuroendocrine-immune regulatory network of Eucommia ulmoides Oliver provides a deep literature review related to chemical compositions, biological roles and pharmacological properties of E. ulmoides and linking them with the NEI associated diseases. The manuscript is well-structured, comprehensive and the literature current and appropriate. In my opinion this paper should be accepted after the minor revision. My main concerns are related to:

  • Table 1: it is huge and so it seems unclear. Please, try to format table in another manner or make it landscape
  • Figure 4: it is necessary to improve the quality of the figure so it could be readable
  • Figure 6 C: the label on x-axis is missing, please add a general label
  • Manuscript is dealing with the biological activity of E. ulmoides, emphasizing the presence of bioactive compounds in its chemical composition. In my opinion the authors should add a special paragraph about antimicrobial and antiviral activity of this plant. Here are some references that could be helpful:

Eucommia ulmoides Flavones as Potential Alternatives to Antibiotic Growth Promoters in a Low-Protein Diet Improve Growth Performance and Intestinal Health in Weaning Piglets 

Daixiu Yuan, Jing Wang, Dingfu Xiao, Jiefeng Li, Yanhong Liu, Bie Tan and Yulong Yin.

Chemical constituents and antimicrobial activity of wood vinegars at different pyrolysis temperature ranges obtained from Eucommia ulmoides Olivers branches, Xiaomei Hou,ab Ling Qiu, Shihai Luo,Kang Kang,Mingqiang Zhu and Yiqing Yao RSC Adv., 2018, 8, 40941–40949.

Eucommia ulmoides Oliver: A Potential Feedstock for Bioactive Products Ming-Qiang Zhu and Run-Cang Sun . Agric. Food Chem. 2018, 66, 22, 5433–5438.

Reviewer 2 Report

First of all, I consider it not an article, but a review, because presents only data from the literature regarding the curative potential of the species  E.ulmoides , being structured similarly to other papers on the same species, for example:

https://www.sciencedirect.com/science/article/pii/S2213453019300412

https://www.ncbi.nlm.nih.gov/pmc/articles/PMC4793136/

The data presented are scientifically correct, but have no degree of novelty, the chemical composition and properties of the species on human health are already known. Moreover, there is already on the market an impressive number of medicines and nutritional supplements containing this plant, their pharmacological effects being presented by the authors in tables 3 and 4, as well as in figure 6. Table 2 and figure 5 show the same thing, I consider it sufficient to present only one of them in the paper. The paper presents a lot of data taken from other studies, has a large number of references, but to ensure the specific scientific soundness of an article, I consider it necessary to present some of the authors' own research. As the range of the species is limited, being native to China, it greatly reduces interest, and clinical trials and advanced research methods are needed to strengthen the chemical and interactive analysis of the active compounds of E. ulmoides Oliv. and elucidate the pharmaceutical mechanism on the neuroendocrine- immune  system

Reviewer 3 Report

The review aims to systematically summarize the chemical compositions, biological roles, and pharmacological properties of E. ulmoides to build a bridge between it and NEI-associated diseases and to provide a perspective for the development of its new clinical applications. 

Generally, the manuscript is interesting, well-written, and organized. Only one concern needs to be addressed before publication: For Figures 3-5: It is necessary to be bigger and clearer for the readers.

Reviewer 4 Report

Dear authors,

your manuscript: "Neuroendocrine-immune regulatory network of Eucommia ulmoides Oliver"  is very well written, clear and consistent. This review is well planned and executed. 

Best regards

Round 2

Reviewer 2 Report

Since many properties and uses of the E.ulmoides are already known, I consider that clinical studies are required to confirm the hypothesis of the authors, and to complete the data presented in the article.